# Attitude Evaluation on Using the Neuromarketing Approach in Social Media: Matching Company's Purposes and Consumer's Benefits for Sustainable Business Growth

**Mihaela Constantinescu** *, **Andreea Orindaru, Andreea Pachitanu, Laura Rosca, Stefan-Claudiu Caescu and Mihai Cristian Orzan**

Marketing Department, The Bucharest University of Economic Studies, 010404 Bucharest, Romania; andreea.orindaru@mk.ase.ro (A.O.); andreea.pachitanu@mk.ase.ro (A.P.); lora.rosca@gmail.com (L.R.); stefan.caescu@mk.ase.ro (S.-C.C.); mihai.orzan@ase.ro (M.C.O.)

* Correspondence: mihaela.constantinescu@mk.ase.ro; Tel.: +40-721-223-896

**Abstract:** The current era has brought about major changes in the way people connect to each other, and social media is a major pillar of this change. In this new communication and connecting environment, companies are under pressure to constantly adapt and become present on platforms where their customers are while being sustainable and profitable in the long run. On the other hand, traditional marketing research challenges have led to the expansion of new research approaches like neuromarketing as a means to gather the most accurate data ever from customers. When referring to how we can use neuromarketing research within the social media context, the current paper aims to fill a gap in the current literature: Using neuromarketing research in social media while taking into consideration both companies' purposes and customers' sought-after benefits. This aim is achieved through two pieces of research that shed light on a model where these purposes are matched with the corresponding benefits, showing the degree of acceptability of four major neuromarketing applications. This model is a stepping stone towards discovering how social media neuromarketing research can be a pillar of sustainable business communication as part of the broader perspective of sustainability in terms of business growth.

**Keywords:** neuromarketing; social media; sustainable communication; quantitative marketing research

## 1. Introduction

In this more-than-ever connected society, social media has gone mainstream as the place where, not only families and friends gather around, but also where customers and companies connect. Within this setting, developing a sustainable model for social media corporate communication is a must for a long-term perspective and life on the market. Testing and evaluating how messages really get across to the customer is a major part of how companies can constantly improve the way their social media corporate communication model delivers the right message to the right customers. In order to reach this goal, one can test new research methods that are likely to give very accurate information about customers' real reactions, which in turn might lead to an even greater level of long-term sustainability of a company's business growth. Taking these two major elements into consideration: Social media usage in corporate communication and developing neuromarketing research to determine social media communication's real impact, this paper aims at bringing science one step further on the pathway towards uncovering the ways we can use neuromarketing research in social media as we attempt

to achieve sustainable business growth. Therefore, this paper aims to add to the conversation on neuromarketing, an area far away from being fully exposed and exploited both in theory and in practice. Additionally, extensive literature scanning proved unsuccessful in discovering other papers approaching a similar topic: Using neuromarketing research within the social media context, thus proving the novelty, originality, and need for the present paper.

## 2. Social Media as Companies' Way of Communicating

Social media represent a new and challenging environment that companies need to adapt to when it comes to communicating with consumers, considering their availability and prevalence. Even if this subject has been fairly debated in recent years, and researchers gave a fair amount of attention to social media as an integral part of marketing communications [1–7], this new landscape of communication created by them has also incurred a new approach in doing business and a new set of business models, which challenge traditional business operations and processes [8].

Although social media are powerful and represent a relevant and important marketing channel, as McIntyre et al. [9] very well pointed out, there is still a fair amount of companies that are "reluctant or unable to develop sustainable and global strategies to allocate resources to engage effectively with social media and their respective supporters". Even if companies have or are striving to have a thorough understanding of how these new communication platforms work, there are still some particularities and communication rules, which social media have introduced that need consideration from their part, as they can enable companies to improve towards a more sustainable and ethical view indirectly, as well as to develop their economic performance [10].

Several authors try to give a general and comprehensive definition of social media, and among the best known is that of Kaplan and Haenlein, who describes them as "a group of Internet-based applications that are built on the ideological and technological foundations of Web 2.0, and that allow the creation and exchange of User-Generated Content (UGC)" [11]. Moreover, Wamba and Carter posit that social media are considered "as disruptive information technology (IT) innovations that have the capabilities of transforming the way we are currently doing business' [12].

According to Statista's Global Digital Population report, there are 4.388 billion active Internet users, and 3.484 billion active social media users worldwide [13], as of July 2019, clearly placing social media tools among the most valuable instruments for companies' marketing and communications strategies, and actually they have become an integral part of the communication strategy of numerous businesses nowadays [5]. Moreover, digital advertising is the fastest-growing category within the global expenditure on media framework [14]. In 2019, the global social penetration rate reached 45%, with North America and East Asia having the highest penetration rate at 70%, followed by Northern Europe at 67% [15].

Social media comes in many forms, including social networks, blogs, microblogs, forums, social gaming, business networks, photo-sharing platforms, or chat apps, just to name a few. Some of the most popular, from the users' point of view, are Facebook (the definite market leader of the social media ecosystem), YouTube, WhatsApp, Facebook Messenger, Instagram, and Twitter [16]. When it comes to social media business statistics, on the other hand, only 20 Fortune 500 companies actually engage with their customers on Facebook, while 83% have a presence on Twitter, but still 91% of retail brands present in the USA use two or more social media channels, while 81% of all small and medium businesses use some kind of social platform [17]. As for Romania, in 2019 the most popular social media platform was Facebook, with 11 million users (potential reach), followed by YouTube with 10 million viewers, and Instagram with 3.8 million users (total advertising audience), while Twitter was only in 7th place, with 379,098 user accounts [18,19]. However, Snapchat has 1.4 million monthly active users, making it a noteworthy platform for the Romanian social media landscape [19]. Considering these figures, the Digital 2019 Romania report, done by we are social and Hootsuite posits that a company can reach 58% of the adults aged 13+ with adverts on Facebook, 22% on Instagram, 16% on LinkedIn, 8.2% on Snapchat, and 2% on Twitter [19]. Moreover, the same report explains that generally,

a typical Romanian Facebook user has the following activity frequency: 1 Facebook page like, 16 posts liked in the last 30 days (all post types), 7 comments made in the last 30 days, 2 Facebook posts shared in the past 30 days, and 17 Facebook adverts clicked on in the last 30 days (any click type) [19], making it clear that thus far Romanian Facebook users would rather create, share, and consume their own or their friends' content than interact with brands. Hence, we considered companies and brands that had a great communication opportunity with their target audience, which was not fully valued at the moment due to the former's lack of an in-depth understanding of the latter's behavior and content needs on social media.

Furthermore, a survey done by Statista in 2016 [20] on the distribution of individuals who use social media platforms in order to share content in Romania, by frequency, revealed that 43% of respondents reported that they used social media for these purposes every day or almost every day, while 16% used it 2 or 3 times a week. What was interesting to note from this survey in terms of user behavior was that 25% of respondents actually said they never used social media for these purposes, implying that they only consumed user-generated content, they did not create it, and what was more, another study revealed that 9 out of 10 Romanians were influenced by social media in terms of trust in brands [21].

According to the Like and Share 2017 [18], a study on social media marketing within Romanian companies conducted by the Valoria company, the efficiency of marketing within social networks was recognized by 80% of companies. In addition, 63% of them said that social media was an effective channel for creating sales opportunities, but only 3 in 10 used social media for sales. Another study published in 2016, "Social media and the Romanian business environment", conducted by EY Romania [22], revealed that 91% of companies considered that using social networks provided them with a competitive advantage, while 74% actually used social networks for promotion, sales, recruitment, or competitive analysis. 58% of the respondent companies spent up to 5 h weekly communicating on social networks, while 20% allocated between 6 and 10 h a week. 36% of respondents said they used social media daily, 28% every 2–3 days, while 23% used it weekly. The study also sheds light on the most important benefits sought by companies when using social networks: Increasing awareness and exposure to the market, while 53% of respondents talked about generating sales opportunities. The research also mentions idea co-generation for product development and converting marketing activities into sales as paramount benefits offered by companies' social media use. When it comes to the type of content offered to followers on social media by companies, the top three includes information about own product and services (77%), information about the company (66%), and information on campaigns, promotions, direct and indirect sales (52%) [22]. One last critical finding of the study was that the majority of the business in Romania that participated in the study intend to consider more strongly the opinion of customers on social media in the future, therefore, confirming the trend whereby companies intended to "normalize their relationship with social consumers who form their buying decisions in the online environment" [22]. This becomes even more important, as the UGC, a huge possibilities provider for marketing efforts [23], is gaining more and more relevance in companies' image among their prospect customers within the social media.

From all this information, one can easily draw the conclusion that, while social media's importance and value is acknowledged by the majority of Romanian companies, they still lag behind when it comes to an understanding on how to gather, analyze and monetize efficiently all the customer data and insights available in this ecosystem, a very important step towards achieving more sustainable communication processes and business models. This is exactly what the present paper aims to bring in the limelight: How communication processes and business models become more sustainable as a result of more accurate information about the customers gathered through new technologies, such as neuromarketing.

## 3. Neuromarketing—What Is It?

Marketing research is, in its essence, about discovering, understanding, and predicting individual behavior on the market [24]. When referring to traditional marketing research tools, research has shown that surveys, for example, are not always filled in with honesty, as subjects tend to obey certain social norms, to maintain a certain outlook [25]. Additionally, evaluating consumers' final decisions (to buy or not to buy) is no use in determining all the underlying paths of consumers' decision-making process [26]. Therefore, humans are growingly considered as a 'black box' that keeps all the secrets of emotions and the decision-making processes, both of them being hardly understood and debugged [27]. Thus, on a background of traditional marketing research not leading to satisfactory enough results for both researchers and business representatives, neuromarketing emerged [28].

The purpose of neuromarketing is to combine neuroscience methods with marketing theories in order to discover the genuine impact of marketing on consumers' behavior, beyond what is visible [29]. More precisely, applying neuroscience techniques in marketing research approaches can create a clearer understanding of the impact that marketing techniques have on consumers [24]. In the end, the aim of neuromarketing is to gain insights that cannot be discovered through other marketing approaches [30], additional to them being considered relatively more objective in combination with traditional behavioral research methods [31]. More specifically, the neuromarketing perspective uncovers emotional engagement as a source for future buying decisions, memory retention, awareness, and attention as the foundation of the future purchase intention [28]. When taking into consideration that approximately 95% of the mental processes are unconscious even to the subject, neuromarketing opens the possibility of getting closer to the invisible part of neuronal connections [27]. Thus, entering the human 'black box' goes beyond just asking customers about their beliefs, feelings, thoughts, memories, or decision-making strategies, as neuromarketing implies studying the neural processes [29], focusing on hidden psychological and biological processes [30]. Moreover, using neuromarketing methods is justified by the fair balance between output and costs, but also by the ability to use it in the early stages of product and brand development with confidence that neuromarketing findings will not be influenced by biases [31].

The first steps towards the occurrence and development of neuromarketing were undertaken in 1999 when Gerry Zaltman from Harvard University conducted the first fMRI research as a marketing tool [32]. Then the 'neuromarketing' concept was created and first defined by professor Ale Smidts in 2002, who stated that it 'designates the use of identification techniques of cerebral mechanisms to understand the consumer's behavior in order to improve marketing strategies' [33]. Neuromarketing defines a practice area that is part of neuroeconomics, which is defined as 'a convergence of psychology, economics, and neuroscience' [34]. The neuroscientific methods used in neuromarketing are as follows according to Lim [29]:

- Electromagnetic: Electroencephalography (EEG): It detects brainwaves changes using an electrodes band or helmet when the subjects are exposed to marketing stimuli;
- Electromagnetic: Magnetoencephalography (MEG): It identifies changes in the magnetic fields generated by electrical brain activity when the subjects are exposed to marketing stimuli;
- Electromagnetic: Steady-state topography (SST): It discovers task-related changes in brain activities with steady-state visually evoked potential (SSVEP);
- Metabolic: Functional magnetic resonance imaging (fMRI): It scans blood oxygenation of the brain generated by underlying neuronal activity;
- Metabolic: Positron emission tomography (PET): It traces radiation pulse to uncover with great accuracy the glucose metabolism inside the brain;
- Electrocardiography (ECG): It measures the heart electrical activity using external skin electrodes;
- Eye-tracking (ET): It measures eye movement and position with the use of dedicated eye trackers;
- Facial electromyography (fEMG): It records the facial muscles and their physiological properties when amplifying tiny electrical impulses;

- Skin conductance (SC): It evaluates the subtle changes in skin conductance responses when the automatic nervous system is activated, testing the extra sweat generated by marketing stimuli;
- Transcranial magnetic stimulation (TMS): It temporarily disrupts specific brain activities in order to observe the marketing stimuli effects on behavior through other methods than evaluating brain activity;
- Neurotransmitter (NT): Implies using various chemical substances that enable transmission of the neurological signals between neurons.

Beyond neuromarketing's benefits in uncovering thus far hidden data, there are various ethical issues raised by approaching humans' unconscious minds and processes. In fact, ethical concerns are one of the most sensitive challenges in applying neuroscience methods in marketing research [29]. For this reason, when developing neuromarketing research approaches, taking into consideration the importance of consent, confidentiality, privacy, and vulnerability of subjects is a premise of neuromarketing acceptance and end results validity [29]. As Hensel et al. [35] point out, the main issues related to neuromarketing include consumer manipulation, lack of transparency, and also shortcomings in consumer autonomy.

Another challenge in using neuromarketing relates to interpreting neuromarketing research projects' findings, which requires extensive knowledge of neuroscience in order to generate accurate and reliable conclusions and recommendations related to the following action plan [29]. Actually, there is a common belief that argues that one ethical issue of neuromarketing is rather related to its usage by marketers or manufacturers rather than scientists developing such research projects [36]. Besides, there is a shortage of sound and detailed methodological guidelines (related to data collection, data analysis, findings interpretation, and conclusions development) when it comes to approaching neuromarketing [29]. This point of view was also confirmed by Lee et al. [37], who proved that in their research through the neuromarketing literature, no methodological primer was found to help newcomers start their neuromarketing research projects. These are especially needed since applying neuromarketing might raise various questions like 'how we manage our (perhaps inevitable) potential for enacting the very realities that we set out to describe, discover and/or deconstruct' [38].

Beyond the methodological novelty of neuromarketing projects, neuromarketing's limitations include high-priced and time-restricted experiments, the need for immovable devices in conducting neuromarketing research, as well as ethical issues that might arise from working with the underlying belief system of test subjects [29]. Beyond a small sample size (unrepresentative for the population in neuromarketing compared with traditional marketing research approaches [32]), a major concern of neuromarketing relates to the issue of reverse inference: Establishing that specific biological reactions are determined by a psychological process [30]. Additionally, neuromarketing challenges also refer to the data it records as it shows changes of brainwaves, eye movement or sweat production, but can give no absolute and individual values for these variables [30]. However, as Stanton et al. [39] point out, one way of tackling this issue might be combining various measures for achieving a more reliable bigger picture.

Given neuromarketing's benefits as well as taking into consideration its limits, neuromarketing is a viable solution for generating sustainable growth. This is proved by neuromarketing's ability to offer the context of more efficient resource allocation in correlation with products and brands being more likely to be accepted on the market [30]. Furthermore, neuromarketing might lead the way towards developing personalized marketing that uses individual-specific stimuli [40]. In the end, the entire purpose of neuromarketing is to find ways to satisfy customer's needs and wants better [25] and hence diminish any waste of resources. Within this framework, the current research aims to uncover the way neuromarketing can be used within social networks, bringing together the perspective of both companies and consumers.

## 4. Using the Neuromarketing Approach in Social Media

In the light of everything above, neuromarketing extends the marketing researcher's arsenal far beyond the boundaries of classical quantitative and qualitative methods, offering additional objective information that, this time, is not only written or spoken, but it is based on analyzing the unconscious response to stimuli. In fact, neuroscience creates a context where research can fully explore how decisions are made among customers and relate to discovering the underlying patterns of attention, attitudes, emotions, and memory [31]. However, the lack of information regarding the usability of such instruments like EEG (electroencephalography), galvanic skin response, or eye-tracking, leads to not considering them as potential ways of exploration for human behavior [24].

Undoubtedly, the fields of academic and scientific research [29] currently use and make the most of eye-tracking, EEG, and GSR for cognitive, developmental, experimental, and media applications in psychology and neurology, but the business environment also has to benefit from the insights of such instruments. For example, social neuromarketing can lead the way in discovering the key message elements that stimulate the right brain areas, which could eventually lead to building more effective advertisements [31]. Understanding the way companies can benefit from the insights given by such instruments is imperative for the sustainable development of the communication strategy [41–43]. The neuromarketing approach becomes even more interesting to explore within the social media communication since digital channels have changed the way research is conducted, shifting the focus on new emerging approaches [14].

Understanding the usefulness of such tools gives businesses access to elaborate information that is more accurate, allowing them to develop strategies and obtain competitive advantages, which in turn can lead to a better positioning in the market and more sustainable growth [44].

Additional to its usage in building communication strategies, neuroimaging techniques might also be used in consumer studies to improve the accuracy of existing classifications of consumer purchasing tendencies [45]. This is achieved through neuromarketing's contribution to gaining a better understanding of the neurophysiological and biological processes that are responsible for decision-making behavior when interacting [30] with different brands and labels.

All of the above can give useful information to both companies and specific researchers who can provide insights on where a user's attention goes and on the time duration, as well as support for the effort to understand and to model user's behavior [46].

Neuromarketing research in social media can assure companies information on how their communication efforts are perceived by the viewers in terms of emotional engagement, memory retention, purchase intention, novelty, awareness, and attention [28]. When consumers interact with their favorite brands, the brain areas responsible for analytic processes are deactivated, and the ones responsible for integrating emotions with buying decisions [47] are being activated. There are studies, based on fMRI screening, which are stating the differences perceived by men and women [48] when interacting with the information provided through social network relations, thus making the segmentation strategies more sustainable and the communication strategies more personalized [49]. Research shows that advertising on Facebook is more effective [50], respondents observed the banner sooner and for a longer period of time, as opposed to other sites, although the location was similar. Because consumers tend to reduce the analytical processes when interacting with their favorite brands [51], this, in turn, might alter their perception over the marketing communication of one company or another, depending on their interest [52].

Another type of neuromarketing research, like EEG, can give companies information about sentiment analysis [53], which can provide data about consumer opinion on their products based on their reactions when interacting with text written in the form of reviews or blogs. EEG techniques can also be used to determine the effectiveness of various marketing communication techniques [54], considering that the emotional response when seeing short movies is stronger than looking at mere pictures.

Other studies indicate that brain scanning and biometrics applied to video, games, retail, and online shopping environments [55], provide information regarding the mental energy invested in decision making.

The field of neuromarketing research is rapidly growing [37], as observed in the numerous analyzed research articles. The neuromarketing field is not limited to the benefits provided for the companies, but also for public institutions and social marketing [31]. Further development in the field is expected, like wearable technology, even fashionable sensors, are gaining a rapid growth within consumers' interests [56], while research projects confirm that neuromarketing research results can estimate marketing communication effectiveness [14] and, therefore, increase the long-term sustainable development of marketing communication of companies.

## 5. Perception on Using Neuromarketing Applications in Online Social Networks

### 5.1. Research Methodology

The present paper corroborates the results of two quantitative researches regarding the perception on using neuromarketing applications in online social networks, one from the organization's perspective and the second from the user's point of view. These two researches were designed together, mirroring each other, thus we can have a real image of the extent to which the organization's purposes meet the user's benefits in terms of collecting and using data from social networks through neuromarketing applications.

Considering that modern companies focus on being as friendly as possible with their customers, trying to avoid a distant and cold relationship based just on transactions, we see the emergence of a need to build a business model that delivers experiences for customers, not just sales for the company and return on investment for shareholders. Customers want to be treated as people, not numbers, thus the use of big data and all the information that is out there should be aligned with the sustainable mindset, in which companies meet their own goals without compromising someone else's wellbeing. Thus, the major question of our research is: Can companies make a correlation between their business purposes when using neuromarketing and the benefits that each customer seeks in social media? This question led us to the idea of double research that evaluated the same thing—neuromarketing usage in social media—from both perspectives (organizations and customers) in order to identify connection points and see how they can be implemented.

Before designing the quantitative researches, we conducted two qualitative researches—an in-depth interview with organizations and a focus group with social network users in order to have a better understanding of the specialized language that can be used when discussing neuromarketing and its use in social media networks. As a result, in the surveys, we have allocated a special section for each of the four most used neuromarketing techniques—eye tracking, face coding, voice recognition, and EEG.

The response to neuromarketing use in online social networks depends on the market's familiarity with both concepts. Considering that the study was conducted on the Romanian market, an ex-communist country, with a relatively new approach to business when compared to the US or Western Europe, it was important to take into consideration this gap when analyzing organizations' and individuals' response to emerging trends such as neuromarketing. At this moment in time, there is not any work dedicated to this precise analysis, only isolated studies on different areas of expertise or consumer segments, such as smartphone adoption process among teenagers [57], technology acceptance (more precisely adoption of Facebook) by Romanian university students [58], UN study on adoption and use of e-Government services: The case of Romania [59], or technology acceptance for marketing strategies [60].

Having this gap of information, we decided that, before discussing the use of neuromarketing in social media, we would take these two concepts and analyze them separately, aiming at first

to understand the place of social media in the life of organizations and individuals, and then the perception on neuromarketing as a general concept.

This research designing process led to the following four objectives for each study, as can be seen in Table 1.

**Table 1.** Research objectives.

| Research Objectives for the Survey with Organizations | Research Objectives for the Survey with Social Media Users |
|---|---|
| 1. The use of social media for organizational purposes (what platforms, with what purpose) | 1. The use of social media for personal reasons (what platforms, with what purpose) |
| 2. The degree of familiarity of the business environment with the concept of neuromarketing | 2. The degree of familiarity of the individual with the concept of neuromarketing |
| 3. The degree of utility for each neuromarketing application in social media | 3. The degree of utility for each neuromarketing application in social media |
| 4. The purposes for which these applications could be used in online social networks | 4. The benefits of allowing the use of these applications in online social networks |

Considering that this type of approach was not found in other papers (putting in the same model the results of two separate studies on neuromarketing use in social media), we started our research design from one of our 2019 working papers [61], in which the authors were testing a model for social media use and education, based on the Technology Acceptance Model (TAM) developed by Davis [62]. TAM was one of the most popular prediction-oriented research models dedicated to predicting the primary motivational factors for the use and acceptance of new technologies and systems [58], thus we were able to adapt it to our research envision and build on the purpose/benefit parallel analysis.

The hypotheses for our study were that, for companies, social media was used as a more personalized way of communication about their products and services, and, for individuals, social media represented both a source of information and a tool of entertainment. Regarding the use of neuromarketing in social media, the hypotheses were that companies were embracing these opportunities, while consumers were more reserved, due to privacy issues.

As a research method, we have chosen the survey, conducting online interviews both with organizations' representatives and social media users. Being exploratory research, we did not focus on representativeness but still tried to keep the sample size and structure as close as possible to a statistically representative study for this specific market. Thus, we interviewed 150 organizations and 385 individual users, and the first selection condition for both categories was to be a social media user.

The sampling structure for the survey with organizations was done according to a set of criteria that influenced both the use of technology and the investments in such activities—capital form (Romanian, foreign, and mixt), area of activity (using the classification of Individual Consumption According to Purpose - COICOP), number of employees (under 9, 10–49, 50–249, 250 and more) and turnover (less or equal to 2 mil. EUR, 2–10 mil. EUR, 10–50 mil. EUR, more than 50 mil. EUR). For the survey with individual users we structured the sample according to age (18–30, 31–40, 41–50, 51–60, more than 60), gender (male, female), and education level (primary school, high school, university).

The questionnaire was built on a Lime Survey platform, and respondents received a link, and responses were automatically sent to the database. Data were collected in the Fall of 2018. For information analysis we used IBM SPSS Statistics 25.

*5.2. Research Results*

In this section, we will present the research results in parallel (organizations and individual users) for each objective, as they are especially designed to be compared, thus showing which of the company's efforts really connected with the market's needs.

5.2.1. The Use of Social Media

This objective must be seen from a business perspective for organizations and from a personal one for social media users. Moreover, these two perspectives must align when putting into practice,

thus that the organization has a sustainable strategy when it comes to its business models and online campaigns.

The use of social media can have different underlying motivators, depending on the platform (Facebook, LinkedIn, Instagram, etc.). Thus, it is important to identify which are the platforms that organizations and individuals use and what activities they associate with each of them. As can be seen in Table 2, both organizations and individuals are most often found on Facebook, be it for business or personal reasons. The second platform, however, differs, with more companies using LinkedIn compared to individuals (58.6% versus 38.1%), whereas Instagram is more popular with individual users (85.1% versus 53.4%).

**Table 2.** Social media platforms used.

| Social Media Platforms Used by Companies | % | Social Media Platforms Used for Personal Interests | % |
|---|---|---|---|
| Facebook | 87.9 | Facebook | 98.5 |
| LinkedIn | 58.6 | Instagram | 85.1 |
| Instagram | 53.4 | Twitter | 20.8 |
| YouTube | 46.5 | Google+ | 43.6 |
| Twitter | 25.8 | LinkedIn | 38.1 |
| Google+ | 25.8 | YouTube | 76.7 |

The foundation of a sustainable communication model is represented by the alignment of organizational purposes to the consumer needs. This also means being where your consumer is and saying/doing what is right for your consumer. Knowing the reasons for which consumers are using social media platforms gives companies the opportunity to adapt and meet their clients halfway. As we can see from Tables 3 and 4, companies do not take into consideration the trends for each platform all the time, and many times fall into the trap of convenience and designing a general communication strategy for several social media networks. For example, Facebook is used by individuals as a tool of communication with other people or for entertainment, whereas companies push products and services promotions on it. For Instagram, a match can be observed, as companies use it for promotions and information, and the individual user is also there for the same reason—following organizations of interest. The most aligned platforms in terms of organizational versus individuals' purposes are Twitter (used for general public information) and LinkedIn (used for recruitment and jobs).

**Table 3.** Purposes for which organizations use social media platforms (%).

| Purpose | Facebook | Instagram | Twitter | Google+ | Linkedin | YouTube |
|---|---|---|---|---|---|---|
| Recruitment | 11.7 | 1.3 | 6.5 | 3.2 | 23.2 | 0.0 |
| Finding business partners | 4.6 | 2.6 | 3.2 | 9.7 | 15.2 | 0.0 |
| Products and services promotion | 23.4 | 33.8 | 22.6 | 25.8 | 17.9 | 38.6 |
| Communicating with clients | 16.8 | 16.9 | 16.1 | 16.1 | 9.8 | 12.3 |
| General public information | 23.4 | 20.8 | 38.7 | 22.6 | 17.0 | 28.1 |
| Gathering information about our market | 8.6 | 7.8 | 9.7 | 16.1 | 10.7 | 5.3 |
| Monitoring the competition | 11.2 | 14.3 | 3.2 | 6.5 | 6.3 | 12.3 |
| Other purposes | 0.5 | 2.6 | 0.0 | 0.0 | 0.0 | 3.5 |

**Table 4.** Purposes for which individuals use social media platforms (%).

| Purpose | Facebook | Instagram | Twitter | Google+ | Linkedin | YouTube |
|---|---|---|---|---|---|---|
| For professional purposes | 2.7 | 1.5 | 6.8 | 40.0 | 36.5 | 2.2 |
| Finding a job | 2.2 | 0.4 | 1.4 | 5.2 | 31.8 | 0.4 |
| Sharing opinions with others, through personal posts or comments on others' posts | 9.0 | 8.1 | 11.0 | 1.7 | 1.2 | 4.0 |
| Communicating with other social media users (messenger) | 17.4 | 11.7 | 1.4 | 6.1 | 2.9 | 1.1 |
| To keep up with what is happening in the world | 11.6 | 10.3 | 28.8 | 16.5 | 4.7 | 15.6 |
| For entertainment | 15.0 | 17.5 | 12.3 | 6.1 | 0.6 | 51.4 |
| To know what my friends/acquaintances are doing | 12.6 | 14.3 | 0.0 | 0.9 | 1.2 | 1.1 |
| To follow people and organizations of interest to me | 11.8 | 14.2 | 27.4 | 7.0 | 16.5 | 17.0 |
| To post photos/videos of moments in my life | 9.4 | 14.6 | 0.0 | 0.9 | 0.0 | 2.5 |
| To meet new people | 6.1 | 6.8 | 5.5 | 1.7 | 4.1 | 2.5 |
| To play | 2.1 | 0.4 | 0.0 | 2.6 | 0.0 | 0.4 |

### 5.2.2. The Degree of Familiarity with the Concept of Neuromarketing

The analysis of neuromarketing utility must be understood in the light of organizations' and individuals' familiarity with the concept. Evidence indicates that people fear change and the unknown [63]. Thus, asking companies to invest in, and people to participate in neuromarketing studies without having, first of all, a real image about the level of their knowledge on the subject is too hypothetical in order to have any meaning.

As can be seen in Figure 1, there is a concentration in three areas when it comes to organizations' familiarity with neuromarketing: Those who know nothing about it, those who declare an average level of knowledge, and a category of respondents who consider themselves familiar with this concept, but do not have the confidence to say that they know too much, hence the high percentage of those who chose a score of 8 out of 10.

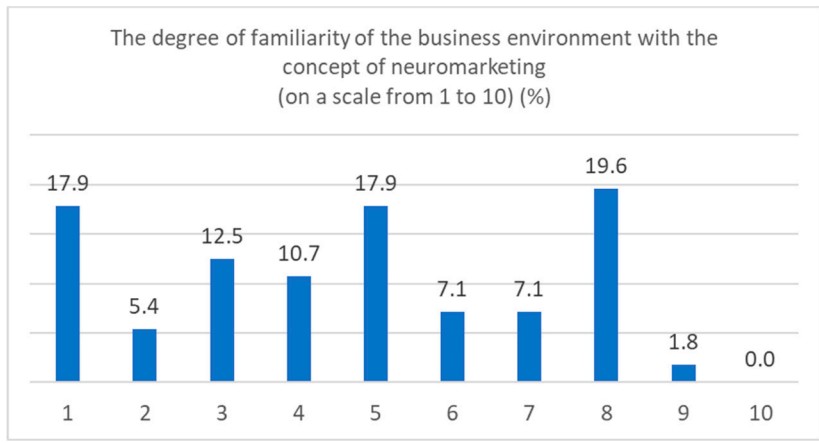

**Figure 1.** Measuring the degree of familiarity of the business environment with the concept of neuromarketing, on a scale from 1 to 10 (%).

The average score of 4.6, on a scale from 1 to 10, showed a relatively low degree of familiarity, which led to the need for more detailed communication on the subject of neuromarketing with the business environment, thus that they can understand both the process of information analyzing, as well as its usefulness. This need was also supported by the mental associations that research participants

made with neuromarketing, where we had a category that thinks of this concept as a technique of influencing/manipulating the consumer. Naturally, there was also a category of organizations that have some correct ideas about neuromarketing (finding the reactions to stimuli, objective research of consumer behavior, analyzing consumer brain waves, etc.), and we can build on this when trying to raise awareness in the market.

The research on individual users showed a concentration of most answers near the minimum limit of the scale (as shown in Figure 2), which denoted a very low level of familiarization of the market with the concept of neuromarketing. This conclusion was also supported by the mental associations that consumers made with this concept, as it should be noted that the highest percentage was for those who did not know what the concept of neuromarketing referred to (16.2%), to which we can add the other 9.2% who have made wrong associations. Besides this quarter of the market that was not at all familiar with the concept, we also have 15.1% that explained neuromarketing as being "something related to marketing". There were, however, also a few correctly made associations with the term neuromarketing—analysis of clients' response to stimuli (9.2%), identifying influence factors (8.1%), analysis of brain activity (7.4%), and analysis of consumer behavior (3.2%). One last thing to mention is the fact that there are also negative associations, in the direction of manipulating the human brain (4.3%).

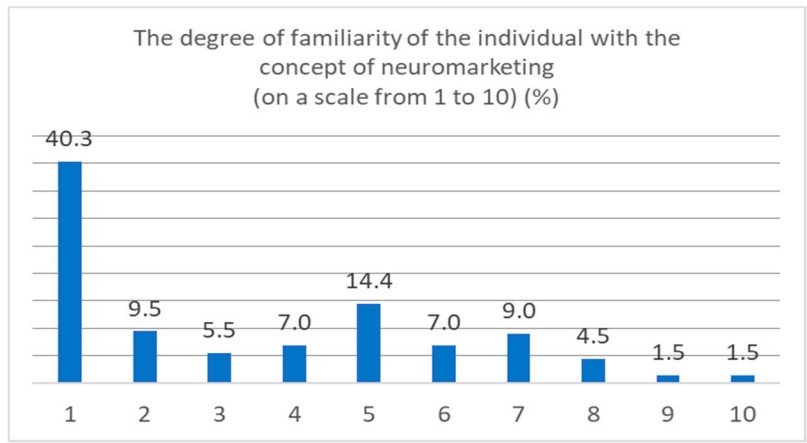

**Figure 2.** Measuring the degree of familiarity of individuals with the concept of neuromarketing, on a scale from 1 to 10 (%).

These conclusions show us that the market is not homogeneous in terms of familiarity with the concept, rather seeing only certain dimensions of it. Having this in mind, we also asked our respondents about the actual participation in neuromarketing studies. When it comes to organizations, only 4% of them have conducted/ordered up until now such studies, and 14.3% of them have used the results of neuromarketing studies done by others. It is encouraging to see that 73.2% of companies from the Romanian market intend to use such research in the future, proving a high level of acceptance on the market.

The research on individuals showed us a percentage of 60.2% of users who would participate in future neuromarketing studies, which again confirms the level of acceptance mentioned above. It is important, however, to know the barriers such as people motivating their refusal by lack of knowledge about the procedure or lack of trust in the data collection process, or how their data will be used after.

5.2.3. The Degree of Utility for Each Neuromarketing Application in Social Media. Purposes and Benefits

This section contains both the results of the third and fourth research objectives, as they are linked—would they use neuromarketing in social networks and why. However, we have to see the answer to the "why" question from two different angles—the business purposes for organizations and

the benefits that users feel they can get. At the end of this part, we will use the results to build a model, which correlates purposes with benefits in order to give organizations the opportunity of a sustainable communication process, showing:

- What are the reasons which have a correspondent in each party;
- What are the business purposes that do not correspond to any benefits in the consumer's view;
- And what are the benefits expected by consumers that do not reflect at all in the present business approach.

Before asking any questions about the four neuromarketing applications evaluated within our research, we have inserted a short description for each of them, thus to make sure that all respondents have a proper understanding of the concept they are evaluating.

In order to find out respondents' take on neuromarketing use in social media (third objective of the research), we used a 5-point scale (Osgood scale), from very useful to very unuseful. The answers to the "why" question (fourth objective of the research) were spontaneous ones, thus we can see how companies and individuals are expressing themselves when it comes to reasoning why neuromarketing in social networks can be a good thing. We have avoided the less insightful option of closed questions, where respondents just agree with the options, which are in front of them, even if those are not the most important ones.

For *eye-tracking*, most companies checked the "very useful" button (as seen in Figure 3A), leading to an average score of 4.02, on a scale from 1 to 5. Consumers, however, tended to be more mellow in their opinion, most of them choosing "useful" (as seen in Figure 3B), which led to a lower average—3.93, but still a good one in terms of meaning for the neuromarketing acceptance level.

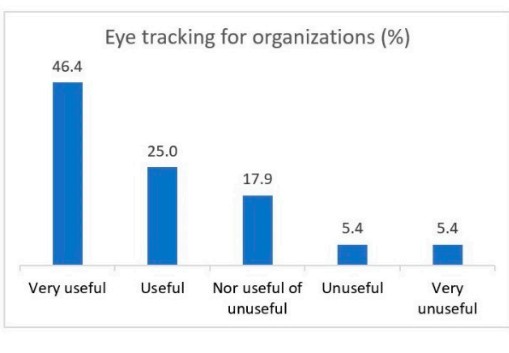
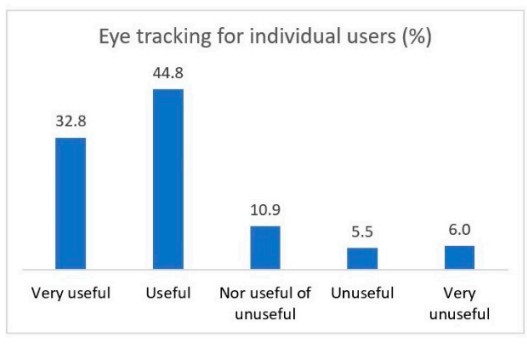

(**a**)                                                                  (**b**)

**Figure 3.** (**a**) Utility of eye-tracking for organizations. (**b**) Utility of eye-tracking for individual users.

The most important purpose for which companies would use eye tracking in social networks is to find out what the elements are that draw people's attention, a fact which also corresponds to the main benefit consumers see in allowing the use of this neuromarketing application for their social media accounts—finding out their areas of interest (Table 5).

**Table 5.** Purpose and benefits of using eye-tracking in social media networks (%).

| Purposes for Which Organizations Would Use Eye Tracking in Social Media Networks | % | Benefits for Which Individuals Would Allow the Use of Eye Tracking in Social Media Networks | % |
|---|---|---|---|
| Find out the area of interest, the elements that draw attention | 30.6 | Find out my interest areas | 52.5 |
| Testing visual promotion elements | 19.4 | It is helpful for companies, not users | 15.2 |
| Finding the right advertisements positioning | 16.7 | Visual rearrangement of a page to make the items of interest more visible | 11.1 |
| Analysis of consumer behavior | 11.1 | Shaping the user/consumer profile | 10.1 |
| Improving the promotion activity | 11.1 | Content efficiency | 4.0 |
| Rearrange/streamline content | 8.3 | To show in the future only what is of interest for the user | 4.0 |

*Face coding* also brings a majority of positive answers from the business perspective, more than $\frac{3}{4}$ found this neuromarketing application useful or very useful, leading to an average score of 4.02 (as seen in Figure 4A). In the consumer survey, face coding gathered most answers in the "useful" option (as seen in Figure 4B), with an average score of 3.76. This score was still a positive one, considering that if people tended to see the usefulness of this application, they would also tend to approve its implementation within social platforms.

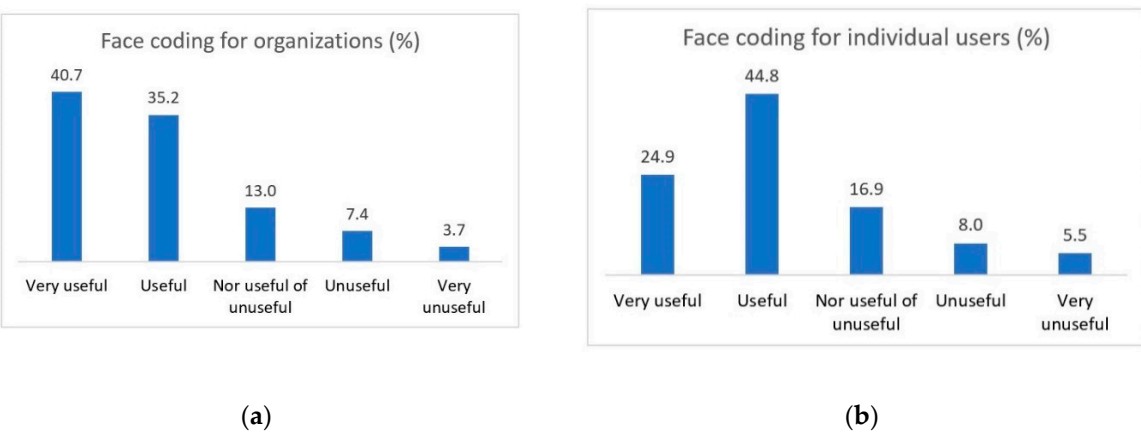

(**a**)　　　　　　　　　　　　　　　　　　　　　　　(**b**)

**Figure 4.** (**a**) Utility of face coding for organizations. (**b**) Utility of face coding for individual users.

More than half of respondents from the business survey declared that they would use face coding in social networks in order to test campaign components or determine the reactions to posted content (as seen in Table 6). The subsequent purpose was corresponding to the most mentioned benefit by individual users—to determine the impact of social media content on users. What is more important is the second benefit—understanding consumer desires and preferences, as they show us individual's need to be understood by companies, thus they can better adapt their social media communication to this.

**Table 6.** Purpose and benefits of using face coding in social media networks (%).

| Purposes for Which Organizations Would Use Face Coding in Social Media Networks | % | Benefits for Which Individuals Would Allow the Use of Face Coding in Social Media Networks | % |
|---|---|---|---|
| Testing campaign components (visual, text, video) | 26.5 | Determining the impact of social media elements on users | 27.8 |
| Determining reactions to the posted content | 26.5 | Understanding consumer desires/preferences | 23.3 |
| Improving the promotion activity | 14.7 | Discovering the true reaction, the one that could be masked by words | 16.7 |
| Measuring users' emotions to posts | 11.8 | Selected content displayed based on user preferences | 13.3 |
| Analysis of consumer behavior | 14.7 | Outline a user/consumer profile | 12.2 |
| To customize the offers | 5.9 | It is helpful for companies, not users | 6.7 |

*Voice recognition* seems to be less useful both in the opinion of companies and individuals (as seen in Figure 5a,b). The average score from business perspective is 3.42 and from individual's perspective is 3.39, which demonstrates a relative indifference to this neuromarketing application.

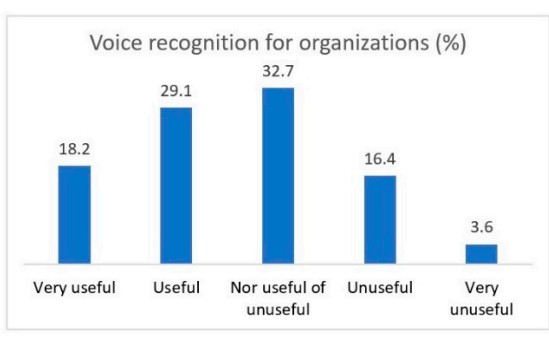

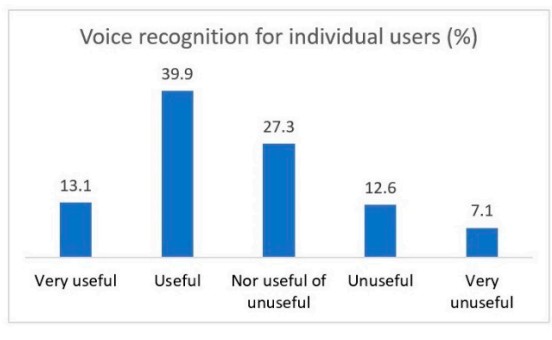

(**a**)                                         (**b**)

**Figure 5.** (**a**) Utility of voice recognition for organizations. (**b**) Utility of voice recognition for individual users.

When asked about the reasons they would use it, companies thought of determining the reactions to posted content as the main purpose (Table 7). The reason which found a correspondent in the most mentioned benefit by individual users was determining the impact of social media elements on users.

**Table 7.** Purpose and benefits of using voice recognition in social media networks (%).

| Purposes for Which Organizations Would Use Voice Recognition in Social Media Networks | % | Benefits for Which Individuals Would Allow the Use of Voice Recognition in Social Media Networks | % |
|---|---|---|---|
| Determining reactions to the posted content | 43.8 | To determine the impact of social media elements on users | 36.2 |
| To create good content for ads | 12.5 | Outline a user/consumer profile | 21.3 |
| Simplifying commands in social media | 12.5 | Understanding consumer desires/preferences | 17.0 |
| In direct discussions with customers | 12.5 | To determine the real reaction, even if it is not described | 12.8 |
| Analysis of consumer behavior | 6.3 | Easier controls on social media, without writing | 4.3 |
| To detect lies | 6.3 | For security of access to the personal account | 4.3 |
| User identification | 6.3 | Choosing the right tone | 2.1 |
| | | It is helpful for companies, not users | 2.1 |

*EEG* (Electroencephalography) was the hardest to sell, as most people saw it as being the most intrusive one. From the individual user's perspective, it was the less useful one out of all the neuromarketing applications evaluated in this study, with an average score of 3.21. The business perspective scored a little bit higher (average score of 3.46), however, both companies and individuals tended to be neutral in respect to the use of EEG in social networks (as seen in Figure 6a,b).

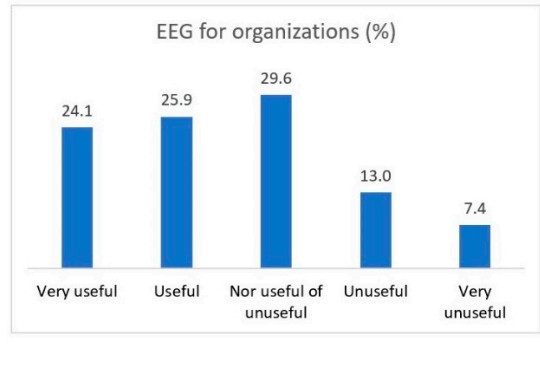
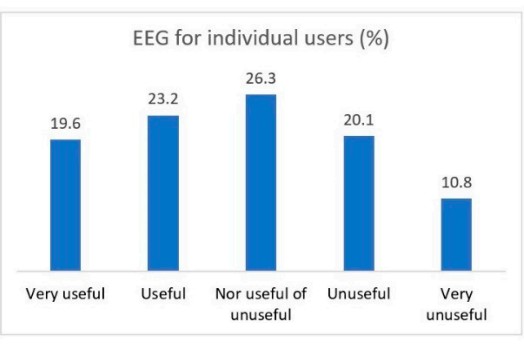

(**a**)                                                          (**b**)

**Figure 6.** (**a**) Utility of EEG for organizations. (**b**) Utility of EEG for individual users.

In the case of EEG, we have good correspondence when it comes to the company's purpose for using it in social networks and individual user's benefit from allowing it to be used—determining the reaction to different stimuli (Table 8).

**Table 8.** Purpose and benefits of using EEG in social media networks (%).

| Purposes for Which Organizations Would Use EEG in Social Media Networks | % | Benefits for Which Individuals Would Allow the Use of EEG in Social Media Networks | % |
|---|---|---|---|
| Determining reactions to the posted content | 45.8 | Identifying the reaction to stimuli, their impact | 31.3 |
| Testing the reaction to different products | 37.5 | Understanding consumer desires/preferences | 29.2 |
| The real reaction, unaltered by the way the individual expresses his opinion | 8.3 | Determining the true reaction, even if it is not described | 25.0 |
| To determine the areas of interest | 4.2 | Outline a user/consumer profile | 8.3 |
| To customize the offers | 4.2 | To determine the right content | 6.3 |

Last, but not least, even if some of the neuromarketing applications now have a lower score in usefulness, their level of acceptance can grow if organizations convert to a sustainable communication model, in which their marketing purposes are aligned with consumer's expected benefit. Thus, the model that we will present below is not just for maintaining what is already working, but also to make social media connections more immersive.

## 6. Purpose/Benefit Model when Integrating Neuromarketing in Social Media

This model brings the two actors of any market face to face—consumers and organizations, with the belief that, if they are in sync, the actions of a company will be reflected in the benefits received by its consumers and then the business model can be sustainable. Considering the topic of our researches, social media use in communication activities, we will limit our analysis to just this part of the business model.

In Figure 7, we also included the average scores for each of the four neuromarketing applications thus that their perceived usefulness can be additionally revealed. This 'purpose versus benefit' model is built from the spontaneous answers given by companies and individuals and taking into account that

we have included in the model only those that have a correspondence in the other party. At the end of this section, we will also comment on the purposes and benefits that do not have a correspondent and how this can affect the sustainability of the communication model.

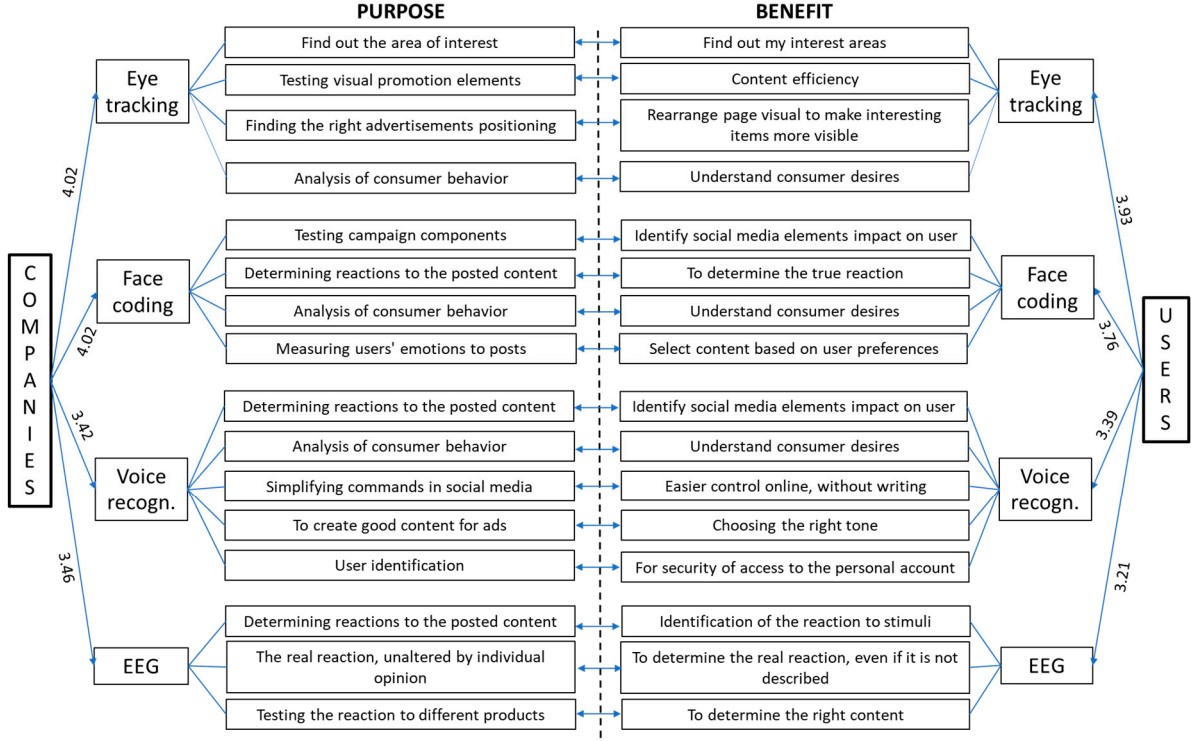

**Figure 7.** Purpose/benefit model when integrating neuromarketing in social media.

The most important part of this purpose/benefit syncing process is to communicate about it, meaning that companies have to explain their purposes in terms of consumers' benefits. The usefulness of our model is that it gives companies the chance to speak the language of their clients, as each business purpose has a corresponding benefit in the users' section. For example, for eye tracking, companies have testing visual promotion elements as one of the purposes. In order for this application to be allowed by users in their social media account, the company has to explain that this testing process is for a greater good-content efficiency, meaning that if it knows the most relevant elements for a user, then future communication targeting that user will be more adapted to his preferences, thus showing him only content that matters.

Besides those elements that are synced in our model, there are also some business purposes or consumers' expected benefits that do not line up:

- Eye-tracking—here we have a statement for one part of the users saying this neuromarketing application was helpful only for companies (15.2%), as seen in Table 5; considering that eye tracking was the most used neuromarketing technique until now, maybe companies should pay more attention to what they say when communicating about the use of such techniques, in order to also make visible the benefits for users;
- Face coding—here we have the same situation, 6.6% of users mentioning that it is an application that is helpful just for companies (Table 6);
- Voice recognition—there are two purposes mentioned by organizations that do not have a correspondence in the users' benefits per se (Table 7)—"it is useful in direct discussions with customers" (12.5%) and "it is used to detect lies" (5.9%); these two aren't and should not be the subject of future communications campaigns, as it may transmit the fact that organizations do not trust their clients, feeling the need to test them;

- EEG—this technique is best known for the fact that it lets companies understand clients' unaltered reactions to stimuli, as the brain doesn't lie; when reading the organizational purposes (Figure 7), we can see that everything revolves around this reaction, but companies should move forward, building on users' reactions and offer a personalized content; this way, the communication will be about what companies can do for their clients, not just about the individual being evaluated over and over; whenever a company communicates about data collection, it has to consider that, in the mind of each person, the first instinct is about protection, especially when it comes to own data. A sustainable approach should emphasize the benefits of allowing data collection.

## 7. Conclusions, Practical Implications, and Future Research Directions

Integrating neuromarketing in social media is not a short-term process for two major reasons. First of all, neuromarketing research requires special equipment and software, and companies interested in going down this road in the future must make investments in devices that are as least intrusive as possible, but, at the same time, capable of capturing all nuances of consumer behavior. In light of this balance that needs to be found when developing neuromarketing research projects, for example, we can use web cameras and microphones available on laptops, computers, tablets, and smartphones, but the data collected using this type of instrument is not of the highest quality.

Secondly, companies should take into consideration users' reluctance to allow data collection about their characteristics and behavior. This is where sustainability steps in, as companies should be very transparent in their business model and build products, services, communication campaigns, etc., in correlation to their customers' needs, maybe even building them together.

The above-mentioned context is not to be seen as a permanent barrier in the process of neuromarketing implementation in social media, but rather as a milestone to be crossed in order to really build that long-term relationship with the customer. The use of neuromarketing allows companies to see through the fog of big data and the curtain of socially desired answers, as it brings to light the real reactions. Thus, the effort has a great reward at the end. However, in order to be sustainable, as this effort is a joint one (companies need the help of consumers for data collection), the communication strategy must focus on showing consumers what the benefits for them are. This is the reason behind the purpose/benefit model when integrating neuromarketing in social media, presented in Figure 7.

One of the most relevant theoretical contributions of this model is the connection between the company's purposes and the user's expected benefits for each neuromarketing technique. In a sustainable mindset, where the organizational objectives are pursued without compromising the user's wellbeing, the communication model is built on properly collected and ethically processed data, with the purpose of offering users a high-quality experience, not just information and conversion.

The empirical implications of this conceptual model can be seen in future validity testing when each purpose-benefit pair should be validated on specific markets. A paramount practical implication of such a model relies on identifying what actions can be taken in order to obtain an increased level of acceptance among users, without which neuromarketing is useless, considering that the data collection process has to first receive the green light from the individuals holding the precious data. If consumers see the benefits and companies' efforts are perceived as real/honest, then social media becomes more adapted, more responsive, and less of an intrusion in the life of its users.

At the end of this section, we also have to address the research's limits as they influence future actions in this direction. The most important limit is the descriptive approach of our analysis, with no correlations between the two major categories of our model—the company's purposes and user's benefits. This limit appears due to the fact that this research is the aggregate result of two separate studies (one on companies and one on consumers), which is limiting our capacity to identify correlation coefficients between them. Being exploratory research, we have assumed this limit from the begging and tried to identify as much information as possible for future causal research.

Future research approaches should take one step further and test in real market situations, the use of neuromarketing applications, as the information presented in this paper is about declared opinions, both for companies and individuals. In terms of the business environment, future researches have to identify what are the most suitable devices to be used in order to deliver and gather the most appropriate data. For users, the research has to focus on how people are getting used to having neuromarketing in their life and identify if this intrusion is changing their behavior. Once this information is tested, companies should validate their communication strategies in the digital world and find this way the best approach for a sustainable business model.

**Author Contributions:** Conceptualization, M.C., S.-C.C., and M.C.O.; data curation, A.P. and L.R.; formal analysis, M.C., A.P., and M.C.O.; funding acquisition, S.C.C. and M.C.O.; investigation, M.C., A.P., and L.R.; methodology, M.C., A.O., and S.-C.C.; project administration, M.C. and A.O.; supervision, M.C. and A.O.; validation, M.C. and A.O.; visualization, M.C., A.O. and L.R.; writing—original draft, M.C., A.O., A.P., and L.R.; writing—review and editing, M.C., A.O., A.P., and S.-C.C.

**Funding:** This work was supported by a grant of the Romanian Ministry of Research and Innovation, UEFISCDI, project number PN-III-P1-1.2-PCCDI-2017-0800/86PCCDI/2018-FutureWeb, within PNCDI III. The APC was also funded by project PN-III-P1-1.2-PCCDI-2017-0800/86PCCDI/2018.

**Conflicts of Interest:** The authors declare no conflict of interest. The funders had no role in the design of the study; in the collection, analyses, or interpretation of data; in the writing of the manuscript, or in the decision to publish the results.

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
