# Peer review of "Attitude Evaluation on Using the Neuromarketing Approach in Social Media: Matching Company’s Purposes and Consumer’s Benefits for Sustainable Business Growth"

_sustainability, doi:10.3390/su11247094_

Round 1
Reviewer 1 Report
First of all, I want to say to the authors that this work present interesting ideas that are potentially of interest to both academics and practitioners. And this is well designed, carefully conducted empirical research for the field of Sustainability.
I find this work interesting and honest. But I would like to suggest some ideas to further improve the manuscript. The included comments must be seen as recommendations to improve the quality of the presented work.
This paper seeks to research how communication processes and business models become more sustainable as a result of more accurate information about the customers gathered through new technologies, such as neuromarketing.
The introduction of the paper delves immediately into the specific the research methodology and sample issues. But this paper must conclude with some theoretical propositions. The paper needs to enhance its theoretical contribution. Agerfalk (2014) may be useful to the authors.
- Ågerfalk, Pär J. (2014), Insufficient Theoretical Contribution: A Conclusive Rationale for Rejection?. European Journal of Information Systems, 23, (6), 593-599.
However, some propositions may be derived from the results.
The review of literature it seems so enough and updated. But this field of research is very important and there are significant applied studies. It may prove useful to deep in the review of the neuromarketing literature or even, in UGC research. I recommend to check the follow works:
Gountas, J., Gountas, S., Ciorciari, J., & Sharma, P. (2019). Looking beyond traditional measures of advertising impact: Using neuroscientific methods to evaluate social marketing messages. Journal of Business Research, 105, 121-135.
Guixeres, J., Bigné, E., Ausín Azofra, J. M., Alcañiz Raya, M., Colomer Granero, A., Fuentes Hurtado, F., & Naranjo Ornedo, V. (2017). Consumer neuroscience-based metrics predict recall, liking and viewing rates in online advertising. Frontiers in psychology, 8, 1808.
Marine-Roig, E., & Clave, S. A. (2015). A method for analysing large-scale UGC data for tourism: Application to the case of Catalonia. In Information and communication technologies in tourism 2015 (pp. 3-17). Springer, Cham.
The research questions are not well developed. It would be necessary to clarify the objectives and argued. On the other hand, I miss the approach of well-argued and well-founded work hypotheses. A case study is developed (but is so local) and it is focused on the peculiarities on the one specific country, Romania. It would be interesting analyse if cultural differences can affect the results.
Finally, in the discussion and conclusion section it would be interesting deep in the limitations. managerial implications and future research lines.
It is remarkable that this conclusion section includes Practical Implications and Future Research Directions
Author Response
|
Comment |
Answer |
|
First of all, I want to say to the authors that this work present interesting ideas that are potentially of interest to both academics and practitioners. And this is well designed, carefully conducted empirical research for the field of Sustainability. I find this work interesting and honest. But I would like to suggest some ideas to further improve the manuscript. The included comments must be seen as recommendations to improve the quality of the presented work. This paper seeks to research how communication processes and business models become more sustainable as a result of more accurate information about the customers gathered through new technologies, such as neuromarketing. |
Thank you so much for your appreciation! |
|
The introduction of the paper delves immediately into the specific the research methodology and sample issues. But this paper must conclude with some theoretical propositions. The paper needs to enhance its theoretical contribution. Agerfalk (2014) may be useful to the authors. - Ågerfalk, Pär J. (2014), Insufficient Theoretical Contribution: A Conclusive Rationale for Rejection?. European Journal of Information Systems, 23, (6), 593-599. However, some propositions may be derived from the results. |
In order to highlight the implications of our model, we have included in the Conclusions section a more detailed presentation for theoretical, empirical and practical contributions, starting from row 610: ‘One of the most relevant theoretical contributions of this model is the connection between company’s purposes and user’s expected benefits for each neuromarketing technique, in a sustainable mindset, where the organizational objectives are pursued without compromising the user’s wellbeing. This way, the communication model is built on properly collected and ethically processed data, with the purpose of offering users a high-quality experience, not just information and conversion. The empirical implications of this conceptual model can be seen in future validity testing, when each purpose-benefit pair should be validated on specific markets. A paramount practical implication of such a model relies on identifying what actions can be taken in order to obtain an increased level of acceptance among users, without which neuromarketing is useless, considering that data collection process has to first receive the green light from the individuals holding the precious data. If consumers see the benefits and companies’ efforts are perceived as real/honest, then social media becomes more adapted, more responsive, and less of an intrusion in the life of its users.’ |
|
The review of literature it seems so enough and updated. But this field of research is very important and there are significant applied studies. It may prove useful to deep in the review of the neuromarketing literature or even, in UGC research. I recommend to check the follow works: Gountas, J., Gountas, S., Ciorciari, J., & Sharma, P. (2019). Looking beyond traditional measures of advertising impact: Using neuroscientific methods to evaluate social marketing messages. Journal of Business Research, 105, 121-135. Guixeres, J., Bigné, E., Ausín Azofra, J. M., Alcañiz Raya, M., Colomer Granero, A., Fuentes Hurtado, F., & Naranjo Ornedo, V. (2017). Consumer neuroscience-based metrics predict recall, liking and viewing rates in online advertising. Frontiers in psychology, 8, 1808. Marine-Roig, E., & Clave, S. A. (2015). A method for analysing large-scale UGC data for tourism: Application to the case of Catalonia. In Information and communication technologies in tourism 2015 (pp. 3-17). Springer, Cham. |
The first additional article recommended was also cited in our paper, but we have extracted a few more ideas from it and integrated in our literature review section.
All in all, we have the following additions to the literature review part of our article: - Current (after all the modifications made in order to comply with the first round of reviews) lines 75-76 become: ‘businesses nowadays [5]. Moreover, digital advertising is the fastest growing category within the global expenditure on media framework [14]. In 2019, the global social penetration rate reached 45%,’ - Current lines 138-140 become: ‘buying decisions in the online environment” [22]. This becomes even more important, as the UGC, a huge possibilities provider for marketing efforts [23], is gaining more and more relevance in companies’ image among their prospect customers within the social media.’ - Current lines 173-174 become: ‘approaches [28], additional to them being considered relatively more objective in combination with traditional behavioral research methods [29]. More specifically, the neuromarketing perspective’ - Current lines 275-277 become: ‘response to stimuli. In fact, neuroscience creates the context where research can fully explore how decision are made among customers and relates to discovering the underlying patterns of attention, attitudes, emotions and memory [31]. However, the lack of information regarding the usability of’ - Current lines 280-302 become: ‘instruments also. For example, social neuromarketing can lead the way in discovering the key message elements that stimulate the right brain areas which eventually can lead to building more effective advertisements [31]. Understanding the way companies can benefit from the insights given by such instruments is imperative for a sustainable development of the communication strategy [41–43]. The neuromarketing approach becomes even more interesting to explore within the social media communication since digital channels have changed the way research is conducted, shifting the focus on new emerging approaches [14].’ - Current lines 344-346 become: ‘consumers’ interests [56], while research projects confirm that neuromarketing research results can estimate marketing communication effectiveness [14] and therefore increase the long-term sustainable development of marketing communication of companies.’
Subsequently, the in-text citations & the references section where modified accordingly. |
|
The research questions are not well developed. It would be necessary to clarify the objectives and argued. |
In order to clarify the need for such a research, we have now included in the revised paper, starting from current line 375, a context presentation and, thus, the major question of our research: ‘Considering that modern companies focus on being as friendly as possible with their customers, trying to avoid a distant and cold relationship based just on transactions, we see the emergence of a need to build a business model that delivers experiences for customers, not just sales for the company and return on investment for shareholders. Customers want to be treated as people, not numbers, thus the use of big data and all the information that is out there should be aligned with the sustainable mindset, in which companies meet their own goals without compromising someone else’s wellbeing. Thus, the major question of our research is: can companies make a correlation between their business purposes when using neuromarketing and the benefits that each customer seeks in social media? This question led us to the idea of a double research that evaluated the same thing – neuromarketing usage in social media – from both perspective (organizations and customers), in order to identify connection points and see how they can be implemented.’ Moreover, in the paragraph starting from row 402, we have already explained why we include in our research those 4 neuromarketing application: we have based our decision on the results of the two previous qualitative researches. From these two researches we have identified also the need of setting two distinct research objectives: one about the use of social media and one about neuromarketing perception: ‘Having this gap of information, we have decided that, before discussing the use of neuromarketing in social media, we have to take these two concepts and analyze them separately, aiming at first to understand the place of social media in the life of organizations and individuals, and then the perception on neuromarketing as a general concept.’ |
|
On the other hand, I miss the approach of well-argued and well-founded work hypotheses. |
In order to offer a more well-argued and well-founded research, we have included, starting from row 415, first of all the explanation behind the research design and afterwards the starting hypotheses: ‘Considering that this type of approach is not found in other papers (putting in the same model the results of two separated studies on neuromarketing use in social media), we have started our research design from one of 2019 working papers [58], in which the authors were testing a model for social media use and education, based on the Technology Acceptance Model (TAM) developed by Davis [57]. TAM is one of the most popular prediction-oriented research models dedicated to predict the primary motivational factors for the use and acceptance of new technologies and systems [58], thus we were able to adapt it to our research envision and build on the purpose/benefit parallel analysis. The hypotheses for our study were that, for companies, social media is used as a more personalized and way of communication about their products and services, and, for individuals, social media represents both a source of information and a tool of entertainment. Regarding the use of neuromarketing in social media, the hypotheses were that companies are embracing these opportunities, while consumers are more reserved, due to privacy issues.’ |
|
A case study is developed (but is so local) and it is focused on the peculiarities on the one specific country, Romania. It would be interesting analyze if cultural differences can affect the results. Finally, in the discussion and conclusion section it would be interesting deep in the limitations. managerial implications and future research lines. |
The reviewer is right, the present case has a local application, more precisely Romania, but we feel that it cannot be yet extrapolated to bigger geographic regions, not before taking into consideration this exploratory research results, that can make us understand better the features of technological acceptance when it comes to something yet so unknown, as it’s neuromarketing. In order to highlight the need for such a research done first of all at the national level, we have included in the revised paper some information about the available literature concerning similar topic, starting from row 322: ‘The response to neuromarketing use in online social networks depends on the market’s familiarity with both concepts. Considering that the study was conducted on the Romanian market, an ex-communist country, with a relatively new approach to business, when compared to US or Western Europe, it is important to take into consideration this gap when analyzing organizations’ and individuals’ response to emerging trends such as neuromarketing. At this moment in time, there isn’t any work dedicated to this precise analysis, only isolated studies on different areas of expertise or consumer segments, such as smartphone adoption process among teenagers [58], technology acceptance (more precisely adoption of Facebook) by Romanian university students [59], UN study on adoption and use of e-Government services: the case of Romania [60], or technology acceptance for marketing strategies [61]. Having this gap of information, we have decided that, before discussing the use of neuromarketing in social media, we have to take these two concepts and analyze them separately, aiming at first to understand the place of social media in the life of organizations and individuals, and then the perception on neuromarketing as a general concept.’ |
|
It is remarkable that this conclusion section includes Practical Implications and Future Research Directions |
Thank you for your kind words! |

Reviewer 2 Report
Dear authors thank you for the opportunity to read your interesting paper.
I would like to comment on several aspects which I believe can improve your manuscript.
The introduction is short and does not present previous studies which were done in this area, nor authors.
Literature review is ok, but is very descriptive. A more sound theoretical debate and theories of neuromarketing could improve your paper.
However, most of my reservations are about your methodology, since it is not theoretically supported. Who else used, employed your method? where is the explanation of the criteria you used? justification.
When was the data collected? which people were chosen? why?
Results, once again they are very descriptive. interesting, but descriptive.
The real contribution is your model purpose/benefit. You should give more emphasis to your contribution in conclusion section.
Small word mistakes in the text (L296 - cu take)
Author Response
|
Comment |
Answer |
|
It is my pleasure to review "The Use of Neuromarketing in Social Media: Matching Company’s Purposes and Consumer’s Benefits for Sustainable Business Growth" for Sustainability. This paper studies the acceptance of using neuromarketing applications in online social networks from the organization’s perspective and also from the user’s point of view, for four neuromarketing techniques – eye tracking, face coding, voice recognition and EEG. While this study provides timely results for the applications of these cutting-edge technologies in business, I do have some major concerns about this paper. |
Thank you! |
|
The title does not describe the contents of the research appropriately, which is using conventional surveys to understand the acceptance of neuromarketing. The research itself does not employ neuromarketing methods. |
The reviewer is right, the previous title may have created the impression that the study is based on neuromarketing research. Thus, we have updated the title of the article from: ‘The use of neuromarketing in social media: matching company’s purposes and consumer’s benefits for sustainable business growth’ to ‘Attitude evaluation on using the neuromarketing approach in social media: matching company’s purposes and consumer’s benefits for sustainable business growth’ |
|
The paper does not provide any basic business information or demographic information about the organizations or the individuals in the study. |
Being an exploratory research, we have tried to keep the sample size and structure as close as possible to a representative study. In order to show this, we have included in the revised paper the following details, starting from row 428: ‘As a research method, we have chosen the survey, conducting online interviews both with organizations’ representatives and social media users. Being an exploratory research, we didn’t focus on representativeness, but still tried to keep the sample size and structure as closer as possible to a statistically representative study for this specific market. Thus, we have interviewed 150 organizations and 385 individual users, and the first selection condition for both categories was to be a social media user. The sampling structure for the survey with organizations was done according to a set of criteria that influence both the use of technology and the investments in such activities – capital form (Romanian, foreign and mixt), area of activity (using COICOP classification), number of employees (under 9, 10-49, 50-249, 250 and more) and turnover (less or equal to 2 mil EUR, 2–10 mil. EUR, 10–50 mil. EUR, more than 50 mil. EUR). For the survey with individual users we have structured the sample according to age (18 – 30, 31 – 40, 41 – 50, 51 – 60, more than 60), gender (male, female) and education level (primary school, high school, university).’ |
|
The analysis is mostly descriptive but do not provide in depth economic/statistical analysis controlling individual effects. |
The reviewer is right. Considering that this is an exploratory research combining the results of two different studies, it lacks the possibility to test the correlation that may appear between company’s purposes and user’s benefits, as the data are in separate data bases. This is one of the major limits of our research, and we have mentioned this in the final section of our paper, with the scope of addressing this limit in future researches, which will surpass the exploratory status and become causal. We have included the following paragraph in the revised paper, starting from row 705: ‘In the end of this section we have to address also research’s limits, as they influence future actions in this direction. The most important limit is the descriptive approach of our analysis, with no correlations between the two major categories of our model – company’s purposes and user’s benefits. This limit appears due to the fact that this research is the aggregate result of two separate studies (one on companies and one on consumers), which is limiting our capacity to identify correlation coefficients between them. Being an exploratory research, we have assumed this limit from the begging and tried to identify as much information as possible for a future causal research.’ |
|
Nevertheless, I think this is an interesting paper addressing an important topic. The paper can be improved if the authors address the comments above. |
Thank you for your appreciation! |

Reviewer 3 Report
It is my pleasure to review "The Use of Neuromarketing in Social Media: Matching Company’s Purposes and Consumer’s Benefits for Sustainable Business Growth" for Sustainability.This paper studies the acceptance of using neuromarketing applications in online social networks from the organization’s perspective and also from the user’s point of view, for four neuromarketing techniques – eye tracking, face coding, voice recognition and EEG. While this study provides timely results for the applications of these cutting-edge technologies in business, I do have some major concerns about this paper.
The title does not describe the contents of the research appropriately, which is using conventional surveys to understand the acceptance of neuromarketing. The research itself does not employ neuromarketing methods. The paper does not provide any basic business information or demographic information about the organizations or the individuals in the study. The analysis are mostly descriptive but do not provide in depth economic/statistical analysis controlling individual effects.Nevertheless I think this is an interesting paper addressing an important topic. The paper can be improved if the authors address the comments above.
Author Response
|
Comment |
Answer |
|
Dear authors thank you for the opportunity to read your interesting paper.
I would like to comment on several aspects which I believe can improve your manuscript. |
Thank you! |
|
The introduction is short and does not present previous studies which were done in this area, nor authors. |
Line 43 from introduction was completed by the following paragraph: ‘media as we attempt to achieve sustainable business growth. Therefore, this paper aims to add to the conversation on neuromarketing, an area far away from being fully exposed and exploited both in theory and in practice. Additionally, extensive literature scanning proved unsuccessful in discovering other papers approaching a similar topic: using neuromarketing research within the social media context, thus proving the novelty, originality and need for the present paper.’ |
|
Literature review is ok, but is very descriptive. A more sound theoretical debate and theories of neuromarketing could improve your paper. |
We have the following additions to the literature review part of our article: - Current lines 275-277 become: ‘response to stimuli. In fact, neuroscience creates the context where research can fully explore how decision are made among customers and relates to discovering the underlying patterns of attention, attitudes, emotions and memory [31]. However, the lack of information regarding the usability of’ - Current lines 300-306 become: ‘instruments also. For example, social neuromarketing can lead the way in discovering the key message elements that stimulate the right brain areas which eventually can lead to building more effective advertisements [31]. Understanding the way companies can benefit from the insights given by such instruments is imperative for a sustainable development of the communication strategy [41–43]. The neuromarketing approach becomes even more interesting to explore within the social media communication since digital channels have changed the way research is conducted, shifting the focus on new emerging approaches [14].’ - Current lines 344-346 become: ‘consumers’ interests [56], while research projects confirm that neuromarketing research results can estimate marketing communication effectiveness [14] and therefore increase the long-term sustainable development of marketing communication of companies.’
Subsequently, the in-text citations & the references section where modified accordingly. |
|
However, most of my reservations are about your methodology, since it is not theoretically supported. Who else used, employed your method? where is the explanation of the criteria you used? justification. |
In order to address this concern, we have included two new paragraphs in the revised paper, in order to present the context before our research and justify the manner in which we have structured research: Starting from row 396: ‘At this moment in time, there isn’t any work dedicated to this precise analysis, only isolated studies on different areas of expertise or consumer segments, such as smartphone adoption process among teenagers [57], technology acceptance (more precisely adoption of Facebook) by Romanian university students [58], UN study on adoption and use of e-Government services: the case of Romania [59], or technology acceptance for marketing strategies [60].’ Starting from row 415: ‘Considering that this type of approach is not found in other papers (putting in the same model the results of two separated studies on neuromarketing use in social media), we have started our research design from one of 2019 working papers [61], in which the authors were testing a model for social media use and education, based on the Technology Acceptance Model (TAM) developed by Davis [62]. TAM is one of the most popular prediction-oriented research models dedicated to predict the primary motivational factors for the use and acceptance of new technologies and systems [58], thus we were able to adapt it to our research envision and build on the purpose/benefit parallel analysis.’ Another element that helped us better design the survey was the qualitative researches conducted before the quantitative ones, from which we could identify which neuromarketing techniques to include in the analysis. This is mentioned in the revised paper starting from row 386: ‘Before designing the quantitative researches, we have conducted two qualitative researches – an in-depth interview with organizations and a focus group with social networks users, in order to have a better understanding of the specialized language that can be used when discussing neuromarketing and its use in social media networks. As a result, in the surveys we have allocated a special section for each of the four most used neuromarketing techniques – eye tracking, face coding, voice recognition and EEG.’ |
|
When was the data collected? which people were chosen? why? |
When was the data collected? Row 441: ‘The questionnaire was built on Lime Survey platform and respondents received a link with it, responses being automatically sent in the data base. Data was collected in the Fall of 2018. For information analysis we have used IBM SPSS Statistics 25.’ How was the sample built? Row 429: ‘Being an exploratory research, we didn’t focus on representativeness, but still tried to keep the sample size and structure as closer as possible to a statistically representative study for this specific market. Thus, we have interviewed 150 organizations and 385 individual users, and the first selection condition for both categories was to be a social media user. The sampling structure for the survey with organizations was done according to a set of criteria that influence both the use of technology and the investments in such activities – capital form (Romanian, foreign and mixt), area of activity (using COICOP classification), number of employees (under 9, 10-49, 50-249, 250 and more) and turnover (less or equal to 2 mil EUR, 2–10 mil. EUR, 10–50 mil. EUR, more than 50 mil. EUR). For the survey with individual users we have structured the sample according to age (18 – 30, 31 – 40, 41 – 50, 51 – 60, more than 60), gender (male, female) and education level (primary school, high school, university).’ |
|
Results, once again they are very descriptive. interesting, but descriptive. |
The reviewer is right. Considering that this is an exploratory research combining the results of two different studies, it lacks the possibility to test the correlation that may appear between company’s purposes and user’s benefits, as the data are in separate data bases. This is one of the major limits of our research, and we have mentioned this in the final section of our paper, with the scope of addressing this limit in future researches, which will surpass the exploratory status and become causal. We have included the following paragraph in the revised paper, starting from row 705: ‘In the end of this section we have to address also the present research’s limits, as they influence future actions in this direction. The most important limit is the descriptive approach of our analysis, with no correlations first of all between the two major categories of our model – company’s purposes and user’s benefits. This limit appears due to the fact that this research is the aggregate result of two separate studies (one on companies and one on consumers), which is limiting our capacity to identify correlation coefficients between them. Being an exploratory research, we have assumed this limit from the begging and tried to identify as much information as possible for a future causal research.’ |
|
The real contribution is your model purpose/benefit. You should give more emphasis to your contribution in conclusion section. |
In order to highlight the implications of our model, we have included in the Conclusions section a more detailed presentation for theoretical, empirical and practical contributions, starting from row 692: ‘One of the most relevant theoretical contributions of this model is the connection between company’s purposes and user’s expected benefits for each neuromarketing technique, in a sustainable mindset, where the organizational objectives are pursued without compromising the user’s wellbeing. This way, the communication model is built on properly collected and ethically processed data, with the purpose of offering users a high-quality experience, not just information and conversion. The empirical implications of this conceptual model can be seen in future validity testing, when each purpose-benefit pair should be validated on specific markets. A paramount practical implication of such a model relies on identifying what actions can be taken in order to obtain an increased level of acceptance among users, without which neuromarketing is useless, considering that data collection process has to first receive the green light from the individuals holding the precious data. If consumers see the benefits and companies’ efforts are perceived as real/honest, then social media becomes more adapted, more responsive, and less of an intrusion in the life of its users.’ |
|
Small word mistakes in the text (L296 - cu take) |
We have corrected the mistake to ‘to take’ |

Round 2
Reviewer 3 Report
The revision has significantly improved from previous version.